# Characterization of Aroma Active Compound Production during Kombucha Fermentation: Towards the Control of Sensory Profiles

**DOI:** 10.3390/foods12081657

**Published:** 2023-04-15

**Authors:** Sarah Suffys, Gaëtan Richard, Clément Burgeon, Pierre-Yves Werrie, Eric Haubruge, Marie-Laure Fauconnier, Dorothée Goffin

**Affiliations:** 1Laboratory of Gastronomic Sciences, Gembloux Agro-Bio Tech, Liège University, 5030 Gembloux, Belgium; 2Laboratory of Chemistry of Natural Molecules, Gembloux Agro-Bio Tech, Liège University, 5030 Gembloux, Belgium

**Keywords:** kombucha, mixed fermentation, volatile organic compounds, stir bar sorptive extraction GC-MS, OAV, flavor, sensory profile

## Abstract

Since the sensorial profile is the cornerstone for the development of kombucha as a beverage with mass market appeal, advanced analytical tools are needed to gain a better understanding of the kinetics of aromatic compounds during the fermentation process to control the sensory profiles of the drink. The kinetics of volatile organic compounds (VOCs) was determined using stir bar sorptive extraction—gas chromatography—mass spectrometry, and odor-active compounds were considered to estimate consumer perception. A total of 87 VOCs were detected in kombucha during the fermentation stages. The synthesis of mainly phenethyl alcohol and isoamyl alcohol probably by *Saccharomyces* genus led to ester formation. Moreover, the terpene synthesis occurring at the beginning of fermentation (Δ-3-carene, α-phellandrene, γ-terpinene, m- and p-cymene) could be related to yeast activity as well. Principal component analysis identified classes that allowed the major variability explanation, which are carboxylic acids, alcohols, and terpenes. The aromatic analysis accounted for 17 aroma-active compounds. These changes in the evolution of VOCs led to flavor variations: from citrus-floral-sweet notes (geraniol and linalool domination), and fermentation brought intense citrus-herbal-lavender-bergamot notes (α-farnesene). Finally, sweet-floral-bready-honey notes dominated the kombucha flavor (2-phenylethanol). As this study allowed to estimate kombucha sensory profiles, an insight for the development of new drinks by controlling the fermentation process was suggested. Such a methodology should allow a better control and optimization of their sensory profile, which could in turn lead to greater consumer acceptance.

## 1. Introduction

Flavor profile monitoring along with other factors, such as the environment in which a food product is consumed, are the most importance factors ensuring consumer acceptance. Indeed, the aroma, which consists of a complex association of volatile organic compounds (VOCs), significantly affects overall consumer experience. Although synergistic interactions can occur between minor constituents of the aroma, consumer perception is mainly impacted by key aromatic compounds, known as odor-active compounds. Monitoring the production and occurrence throughout food processing is therefore of crucial importance [1].

Fermentation has regained attention in the food industry as a natural, cost-effective, versatile, and proven technology that extends the shelf-life of food products and enhances their nutritional content while developing foods with interesting sensory profiles [2]. Fermentation can be applied to produce natural drinks to replace conventional soft drinks.

In this context, Kombucha (from the Japanese word for “Tea Algae”) is a fermented drink originating from China, Korea, and Japan (220 BC). It is popular for its detoxifying, antioxidant, and energizing properties, and is also recognized as active against digestive problems [3,4]. This refreshing drink is obtained through the fermentation of sweetened tea, caused by the action of a “Symbiotic Culture Of Bacteria and Yeasts” [5], named SCOBY. The latter can also refer to the cellulosic film produced by specific bacteria on the surface of kombucha during fermentation [6]. The presence of specific strains of microorganisms (MOs) working as a symbiotic culture during fermentation prevents the contamination and growth of spoilage and pathogenic microorganisms [7]. Following inoculation of sweetened tea with a SCOBY and its liquid starter (fermented liquid produced by the previous kombucha batch [8]), the yeasts hydrolyze sucrose into glucose and fructose, which are in turn converted to ethanol as the main metabolite [9,10,11]. This alcohol is then used by the acetic acid bacteria (AAB) as a substrate to produce acetic acid via oxidation, leading to low levels of alcohol, although kombucha is considered a non-alcoholic beverage [12]. Additionally, AAB converts glucose to gluconic acid and cellulose (SCOBY), and fructose to acetic acid [6]. Moreover, the well-known [4,13,14,15] therapeutic effects of kombucha derive from its chemical composition, and mainly from the tea polyphenols activated and secondary metabolites produced during fermentation [13]. In aerobic fermentation conditions, a slightly carbonated and acidic refreshing drink is obtained, containing several organic acids, amino-acids, vitamins, tea polyphenols, and some hydrolytic enzymes [16].

Multiple factors affect the concentration of the different constituents present in non-alcoholic fermented beverages. Thus, the initial matrix and sugar contents [17], the fermentation duration [18], the incubation temperature [19], and the nature of the kombucha culture [20] mostly influence the concentration of the components of this complex beverage [21]. Lončar et al. (2006) point out that the fermentation time is the most influential parameter on the composition of the fermentation products, followed by temperature and the concentration of the liquid starter [3].

To date, the most prevalent and abundant SCOBY taxa are the yeast genus *Brettanomyces* and the bacterial genus *Komagataeibacter* (formerly *Gluconobacter*), sampling the upper and lower SCOBY layers, respectively [22]. Amongst the AAB identified by culture-dependent studies [23], *Acetobacter xylinum* (reclassified as *Gluconacetobacter xylinus* [24]) is considered as the most efficient cellulose producer in kombucha [25]. While *Acetobacter* species generally dominate [26,27], the genera *Gluconacetobacter* and *Lactobacillus* have occasionally been isolated [28]. The literature is inconsistent about the presence of lactic acid bacteria (LAB) in kombucha, as they are sometimes found in low concentrations [29,30] and simply not reported in other cases [7]. Moreover, fungal genera have also been identified, mainly *Zygosaccharomyces*, *Saccharomyces*, *Dekkera*, *Pichia* [16,31], *Lachancea,* and *Starmerella* [22].

Thus, the composition (organic acids, tea polyphenols, catechins, etc.), the physicochemical properties, and the aromatic compounds content in kombucha are not only dictated by the microorganisms present but also by the pathways they use during the fermentation process, all having a positive impact on the final product flavor [32,33].

The aromatic compounds in kombucha can be divided into three categories according to their origin: native from the used matrix (tea constituents and derived ingredients), from the saccharide source substrates, and from the metabolites produced by the microorganisms during fermentation [34]. The composition and concentration of VOCs vary according to the fermentation stages of the kombucha as well as the fermentation temperature [35]. Furthermore, the production of sapid metabolites such as organic acids during fermentation also adds to the overall sensorial experience [36]. The main VOCs reported in the literature are classified into six different families, namely carboxylic acids, alcohols, aldehydes, ketones, esters, and benzenoids [34].

Since the sensorial profile is the cornerstone for the development of kombucha as a beverage with mass market appeal, advanced analytical tools are needed to gain a better understanding of the kinetics of aromatic compounds during the fermentation process to control the sensory profiles of the drink.

That is why in this study, after analyzing the physico-chemical parameters of the beverage as a function of the fermentation key parameters (such as fermentation time and temperature for instance), the kinetics of volatile organic compounds was determined using stir bar sorptive extraction—gas chromatography—mass spectrometry (SBSE−GC−MS), which allows an optimal capture of volatile and semi-volatile compounds in aqueous matrices [37,38]. The correlation with microbial metabolic pathways was then established.

As mentioned earlier, although kombucha presents a complex mixture of VOCs, only a part of them contribute to the overall aroma and influence consumer perception. Ortho-nasal detection odor thresholds were thus considered to determine odor-active compounds and evaluate consumer perception of kombucha. Eventually, this methodology will result in—by understanding the main fermentation parameters impacts on the kombucha process—the designing of new drinks by controlling the fermentation process to optimize their aromatic profile for gaining consumer acceptance.

## 2. Materials and Methods

### 2.1. Materials and Chemical Reagents

The Chun Mee Moon Palace green tea and Lien Son black tea leaves used in this study were sourced from China (Zheijiang) and Vietnam (Yen Bai), respectively, and were certified to be organically grown (limiting the probability of undesirable molecules in the beverage). Sucrose used for the fermentation was a beet refined white sugar from Tirlemont (Belgium). This sucrose type avoided the development of any additional taste and allowed to increase the alcohol level by favoring the yeast’s activity during the fermentation [4]. The culture medium used for tea fermentation was a symbiotic culture of bacteria and yeast (SCOBY) from Fairment (Berlin, Germany). This cellulose biofilm was preserved in kombucha. Enzytec enzyme kits for the determination of sugar, ethanol and acid content of the samples were obtained from R-Biofarm (Pfungstadt, Germany). The solvents used in this study were of analytical quality.

### 2.2. Kombucha Preparation

The tea was prepared by infusing Chun Mee Moon Palace green tea leaves and Lien Son black tea (1:1) with a total concentration of 8 g/L in mineral water previously heated to 75 °C [35]. After infusing for 10 min, the tea solutions, made in three repetitions, were filtered and sweetened with white beet sugar (60 g/L) [3,35]. Once the solutions were cooled (<35 °C), the culture medium composed of SCOBY and its liquid starter was added at a concentration of 10% (*v*/*v*). The containers were covered with a gauze held by an elastic band (aerobic conditions) and then placed in the oven (Memmert UF55) for fermentation for 14 days at 20 and 30 °C, respectively [16]. Measurements of the kinetics were taken on day 0, 2, 4, 7, 9, 11, and 14 of fermentation (referred to as D0, D2, D4, D7, D9, D11, and D14 later in the manuscript). Each day of measurement, a volume of 10 mL was sampled and placed at −20 °C to carry out the different determinations and to evaluate the fermentation kinetics.

### 2.3. Dynamic Analysis of the Physico-Chemical Parameters of the Drink

The pH was measured with a pH meter (Hach BeRight; SensION+ pH1; S/N 605069; IP67, Hach Company, CO, USA). The pH during the fermentation was monitored via direct measurement in the fermentation medium every sampling day.

Quantification of the major components, namely ethanol (EtOH), carbohydrates (glucose, fructose, sucrose), and organic acids (acetic and lactic), was performed with spectrophotometry (Molecular devices SpectraMax ABSPlus; S/N ABP00716, LLC, San Jose, USA) using colorimetric enzyme kits (R-Biopharm, Darmstadt, Germany). For these quantifications, the sample was diluted in 200 mM pH 7 phosphate buffers (except for lactic acid where the phosphate buffer was at pH 8). These assays were performed on the samples collected to obtain the kinetics (production/conversion) of these molecules of interest. Ethanol determination was performed at the end of the fermentation process to ensure that the levels are below 1.2%, per the European legislation (1169/2011) (0.5% for non-EU countries [39]) for being considered as a non-alcoholic drink [12].

### 2.4. Chromatographic Analysis of the Development of Volatile Organic Compounds

Volatile organic compounds were characterized using GC-MS [38,40,41]. Beforehand, VOCs were sampled via sorption on a magnetic bar (SBSE Gerstel-Twister 10 × 0.5 mm) coated with polydimethylsiloxane (PDMS) as the adsorption resin [42]. The magnetic bars were conditioned in 10 mL of a solution of ACN/H2O (50/50 *v*/*v*) under magnetic stirring for 2 h; then, the magnetic bars were rinsed with distilled water and dried under nitrogen flow for 2 h at 260 °C (initial temperature of 40 °C, flow of 10°/min, pressure of 2 bars) in a thermo-desorption tube. For the samples analysis, they were thawed in a water bath (30 °C) and homogenized; a 5 mL sample was transferred into a vial, and NaCl was then added to reach a concentration of 20% (*w*/*v*). A conditioned SBSE was then placed in each vial and agitated with a magnetic stirrer for 2 h. The magnetic bar was then rinsed with distilled water and partially dried with absorbent paper. A total of 1 µL of heptan-1-ol at a concentration of 100 mg/L (in ethanol) was added to each sample as the internal standard. VOC analysis was performed using gas chromatography (GC Agilent Technologies 7890A) coupled with mass spectrometry (5975C). The injection was performed in the split-less mode with a flow rate of 1.2 mL/min, using helium as the carrier gas. The column used was a Carbowax 20 M, with dimensions 30 m × 250 µm × 0.25 µm. The temperature gradient of the oven was as follows: 40 °C for 2 min, then 6 °C/min until 300 °C for 5 min. The total analysis time was 50 min. Retention indices (RIs) were calculated based on a mixture of n-alkanes between C_6_–C_30_ (Sigma Aldrich, Darmstadt, Germany) [43,44]. The chromatograms of the samples were analyzed with the Mass Hunter Analysis software (B.08.00 software, Agilent Technologies, Santa Clara, California, USA), and the identification of the VOCs was performed by comparison to the NIST17 and WILEY275 spectral libraries (National Institute of Standards and Technology, Gaithersburg, USA) and RI from the literature [45]. To optimize compound identification and data processing, the following filters were applied as a first rough compound sorting on the raw data: match factor (percentage of probability of correct identification of the compound under consideration) ≥70% and maximum area under the curve ≥0.05% (except for aroma analysis: ≥1%). Next, a manually disciplined treatment approach consisted in the validation of all molecules one by one, referring to the mass spectrum by consulting the spectra of reference molecules in the literature. Additionally, RI was finally considered to match the validation hypothesis [41].

The semi-quantification of volatile compounds was calculated relative to the injected internal standard. The determination of OAV (odor-activity value) was calculated as a function of the considered compound concentration in the sample divided by its orthonasal detection perception threshold in water, referred in the literature [46]. This concept allowed to study the contribution of the compounds to the aroma of the drink [47].

### 2.5. Raw Data Analysis

All chemical analyses and fermentation medium measurements were performed in three replicates, and the values were expressed as means (n = 3) ± standard deviation (SD). Basic statistical data processing was performed using GraphPad Prism (8.0.1 software, GraphPad software Corporation, San Diego, California, USA) and Microsoft office Excel (Excel 2021 software, Microsoft Corporation, Redmond, Washington, USA) software. Student’s *t*-tests were performed on the pH and the spectrophotometry data to determine if there were significant differences (n = 3; α = 5%) measured between the stages of fermentation.

Principal component analysis (PCA) using the Spearman correlation matrix was conducted on mean VOC class concentrations, considering kombucha fermentation stages as variables. PCAs were performed on all VOC concentrations, considered as individuals. Kombucha fermentation stages were represented as vectors. Principal components PC1 (Dim1) and PC2 (Dim2) were considered and reported in this article. Variables representation quality was illustrated as cos² values, rated by contribution scale. PCAs were performed in R (R 4.0.2 software, R Development Core Team, Boston, United States).

## 3. Results and Discussions

The set of experiments carried out in this study included the fermentation of kombucha at two different temperatures: 20 and 30 °C. The choice to present and discuss only the results obtained under the 30 °C temperature modality was made considering the similar trends observed in the results between the two temperatures implemented. Moreover, 30 °C temperature modality induced the most optimal fermentation from the development point of view of the various organic acids [29,34,35] not to mention that the controlled fermentation time and the set liquid starter concentration were indicators influencing the beverage [3]. However, as the fermentation temperature might induce variations in the VOC synthesis and ultimately on the aroma profile of the drink, an insight on the temperature impact is proposed in this manuscript (by considering the modality of 20 °C), regarding Section 3.2.2.

### 3.1. Physico-Chemical Characterization of Kombucha during Fermentation

#### 3.1.1. Evolution of pH during Fermentation

As shown in Figure 1, the initial pH was 4.05 ± 0.00 and decreased to 3.00 ± 0.03 after 14 days of fermentation. The evolution of the pH decreases linearly and constantly throughout the fermentation, although very high significant differences (*p* < 0.0001) are measured all along the fermentation.

The study by Zhen-ju Zhao et al. (2018) focusing on kombucha fermentation also describes a decrease in pH during fermentation [7,26,34,36,48]. In addition, the sequencing of a rapid decrease in pH at the beginning of fermentation (0–10 days) followed by a slower decrease at the end (10–14 days) has been demonstrated [9,34]. Indeed, both parameters are indirect evidence of the growth of MOs present in the consortium and the accumulation of flavor metabolic products [9,16,19,34].

#### 3.1.2. Carbohydrate, Alcohol, and Acid Concentration Kinetics during Kombucha Fermentation

The evolution of the concentrations of carbohydrates, alcohol, and organic acids during kombucha fermentation is presented in Figure 2.

First, the sucrose concentration decreases in a very highly significant way (*p* < 0.0001) over time (from 57.4 g/L ± 0.09 at D0 to 25.2 ± 2.85 g/L at D14), with a highly significant decrease (*p* < 0.01) in sucrose content between D7 (44.2 ± 3.92 g/L) and D11 (24.9 ± 2.22 g/L). The amounts of residual sucrose are in accordance with those reported from previous studies on black and green tea kombuchas [17,34]. Degradation kinetics of this disaccharide by the yeast taxa into glucose and fructose units through the glycolysis pathway [49] allows the assimilation of simple sugars by them [7,50]. As a result, since the concentration of simple sugars is inversely proportional to that of sucrose, it increases during fermentation [3]. The concentration of glucose (from 0.40 ± 0.21 g/L at D0 to 14.0 ± 1.38 g/L at D14) in the fermentative medium is higher than that of the fructose (from 0.93 ± 0.24 g/L at D0 to 6.45 ± 2.03 g/L at D14), increasing in a very highly (*p* < 0.0001) and a highly (*p* < 0.01) significant way over time, respectively. In fact, during the hydrolysis of sucrose, yeasts convert fructose and glucose simultaneously to ethanol and CO_2_ on the one hand, with a more rapid utilization of fructose as compared to glucose [7,9]. On the other hand, the AAB preferential carbon source is glucose, principally for SCOBY, resulting in cellulose synthesis [9,34,48,49,50,51].

Ethanol evolution increases in a highly significant way (*p* < 0.01) during the fermentation process, starting from 0.13 ± 0.02% at the initial stage until 0.74 ± 0.14% at the final stage. Ethanol synthesis at the beginning of fermentation is a trend observed in the literature [7,52], correlated with an exponential increase in the yeasts population growth [17,34]. Previous studies reported a final ethanol content of approximately 0.5% [36] and 0.6% [35] after 12 and 20 days of fermentation, respectively (for initial sucrose contents of 50 and 100 g/L, respectively). The kinetics of ethanol production can be explained by the initiation of *Saccharomyces* genus to the yeasts metabolic activity, the first actors in the fermentation, which synthesize ethanol thanks to the availability of carbonaceous and nitrogenous substrates as well as oxygen in the medium [7,50]. In addition, these low levels of ethanol in the samples at the end of fermentation are below the 1.2% (*v*/*v*) per the European regulation (1169/2011) for considering it as an non-alcoholic drink [12].

Moving to the composition of the kombucha microbiome provided by the liquid starter and the kombucha SCOBY, the ethanol produced by the yeasts is converted via various fermentative pathways into organic acids by different bacteria [16,20,48,49,50]. Therefore, among these organic acids, the kinetics of acetic acid production are staggered in time to that of ethanol: starting from 0.63 ± 0.07 g/L at the initial stage to reach 8.42 ± 1.03 g/L at the final stage (*p* < 0.001). Studies reported the start of acetic acid production at D3 [51] and an exponential increase in bacteria population growth from this stage [34].

Despite the very low values measured considering lactic acid synthesis during the fermentation process, it increases significantly, ranging from 0.05 to 0.06 ± 0.00 g/L for the initial and final stages, respectively (*p* < 0.05). Indeed, LAB represent a minority share of the microbiome present in the kombucha SCOBY compared to AAB [19,22]; lactic acid is therefore less produced compared to acetic acid.

Once the organic acids have been synthesized by the microbiome, the inhibition of some and the evolution of others towards the synthesis of aromas allows the production and evolution of these acids to be highlighted.

### 3.2. Study of the Development Kinetics of Volatile Organic Compounds

#### 3.2.1. Chemical Analysis of Major VOCs and Metabolic Pathways Involved in Fermentation

The evolution of VOC content in the raw matrix and during the kombucha fermentation process is shown in Table 1. A total of 87 VOCs were detected in kombucha during the fermentation stages, including 28 carboxylic acids, 23 alcohols, 10 ketones, 9 terpenes, 7 esters, 7 aldehydes, 2 phenols, and 1 benzene. In fact, the SBSE technique tends to capture VOCs and semi-VOCs from both the aqueous and gaseous phase of the matrix [37,38]. In this case, sensitivity is increased as compared to routine extraction techniques such as SPME or HS-SPME [34,37].

From a global point of view, all reported classes (except for esters and phenols) are found in the raw matrix. Indeed, the fermentation process changes dynamically the VOCs tea profile into a different mixture of compounds [7,33,34]. These families of molecules are mainly metabolites synthesized by the microbiome on the basis of organic substrates present in the environment [16,36]. Metabolic pathways relative to kombucha fermentation are adapted to the nutrients present in the tea (mainly nitrogenous compounds) [17,35].

Classes found in the tea matrix globally tend to decrease during fermentation (namely carboxylic acids, aldehydes, benzenes, ketones, and terpenes). Fatty acids are a specific type of carboxylic acids in the tea matrix [36,53,54]. The compounds found in the highest concentrations are hexadecanoic acid (568.52 ppm and 427.35 ppm in black and green teas, respectively), (E)-hexadec-9-enoic acid (226.30 ppm and 129.69 ppm in black and green teas, respectively), and pentadecanoic acid (150.78 ppm and 102.42 ppm in black and green teas, respectively). Total concentrations of carboxylic acids are 1222.65 ppm and 994.75 ppm for black and green tea, respectively.

Terpenes are also mainly present in tea (representing a total concentration of 635.88 ppm and 669.93 ppm for black and green teas, respectively) with major compounds such as squalene (563.30 ppm and 510.25 ppm in black and green teas, respectively) and limonene (60.22 ppm and 121.01 ppm in black and green teas, respectively). Terpenes present diverse properties involved in chemical resistance of tea plant against biotic and abiotic stresses [55,56]. More precisely, squalene is a natural lipid of the triterpene class involved in the synthesis of cholesterol, steroid hormones, and vitamin D [57]. Limonene has a fruity, acidic aroma and is known for its antiseptic, antiviral, and sedative properties [58].

The synthesis of new compounds during the fermentation is also reported [7,34,36] (both introducing new compounds into existing classes (carboxylic acids and alcohols) and new full-fledged classes (esters and phenols)). Moreover, carboxylic acids are also the major components of the fermentation process, followed by alcohols. The use of these two classes of molecules as a basis for the formation of other types of compounds such as esters [59], phenols, and terpenes mainly (molecules formed using aldehydes and ketones) is reported in the literature [34,60].

Carboxylic acids mostly found during the kombucha fermentation are decanoic, hexadecanoic, hexanoic, and octanoic acids, synthetized during fermentation process to reach final concentrations of 89.08 ± 7.94, 47.75 ± 12.37, 51.37 ± 9.61, and 303.54 ± 9.94 ppm, respectively. These major compounds are reported in previous studies [34]. Total carboxylic acid content at the end of fermentation is 653.95 ppm. Major alcohols metabolized during fermentation are 2-phenylethanol and 3-methylbutan-1-ol, reaching 273.15 ± 2.16 and 113.39 ± 8.02 ppm at the final fermentation stage (for a total alcohol content of 484.51 ppm).

Alcohols found in kombucha are mainly metabolites of *Saccharomyces* yeasts [51,60,61]. More precisely, mixed fermentation allows the production of enriched VOCs profiles by non-*Saccharomyces* yeast strains such as *Zygosaccharomyces sp.*, *Dekkera sp.*, *Candida sp.,* and *Pichia sp.* [13,22,33]. They contribute to the final aroma of the product but also to the formation of organic acids by their oxidation [1,61]. Together with aldehydes principally, they are the main substrates for the formation of organic acids and esters by the metabolic pathways of the bacteria present in the SCOBY [34]. In other words, alcohols are used by the MOs to synthesize aromatic volatile acids that acidify the medium [36,60].

Intermediary classes such as aldehydes, benzenes, and ketones are present in the raw matrix and used during the fermentation for the synthesis of both microorganisms’ secondary metabolites and aromatic molecules [7]. Indeed, their contents at final stage are relatively low (6.37 ppm for aldehydes, 11.33 for ketones) or not detected (benzenes). Ketones are metabolized by yeasts mainly through the oxidation of alcohols [60].

Now considering classes mainly formed during the fermentation process, ester and phenol total contents represent 71.71 ppm and 54.06 ppm, respectively. Final terpene content decreased to 78.45 ppm. However, new compound synthesis during the fermentation process such as Δ-3-carene, α-phellandrene, γ-terpinene, and m- and p-cymene occurs.

The evolution of content of different classes during the fermentation process is illustrated in Figure 3. Trends outlined previously concerning classes kinetics are confirmed using the Spearman correlation established between the evolution of VOC classes during fermentation stages (Figure 4). Indeed, classes such as ketones are negatively correlated with carboxylic acids and phenols; aldehydes are negatively correlated with alcohols and esters; phenols are positively correlated with esters and carboxylic acids.

In addition, total content of released molecules as a function of fermentation stages can be highlighted. Fermentation increases total VOC content over time, with a maximum of 1790.92± ppm at D4. Indeed, during the fermentation process, both VOC composition and relative content fluctuate [34,36].

In this case, the association of yeasts and AAB drives the fermentation process and mainly contributes to the chemical composition of kombucha [7]. Essentially, through the MO activity, reagents, and substrates from the fermentation media (raw matrix, carbonaceous substrate and kombucha broth) result in the formation of metabolic and aromatic products. As described previously, most known and studied metabolic pathways driving the fermentation process are glycolysis pathway allowing yeasts to degrade carbonaceous substrates into ethanol, products then implied for the synthesis of organic acids and cellulose considering AAB pathways [49].

As the SCOBY is a complex consortium composed of diverse populations from both yeast and bacteria taxa, complementary metabolic pathways might be used (not to mention inter-species interactions), leading to the synthesis of these specific VOC profiles (Table 1). As a result, the correlation between the most synthetized VOCs and MO consortium evolution during fermentation could provide a better understanding of this fermentation process. The synthesis of phenethyl alcohol and isoamyl alcohol by *Saccharomyces* genus (sometimes by *Candida* and *Lactobacillus* genus) leading to ester formation (ethyl acetate main structure) is reported in the literature [62]. Moreover, terpene synthesis occurring at the beginning of fermentation could be crossed with the yeast activity as well [63]. Recent research on engineered yeast used for the microbial synthesis of terpenes was conducted, linked with their high mevalonic acid metabolic flux and their important intracellular acetyl-CoA, enzymes, and substrates concentrations [64]. The study of Ferremi Leali et al. (2022) demonstrated inter-species metabolic interactions through the improved VOC synthesis diversity profile (production of geraniol, Δ-3-carene, and p-cymene) in co-cultures consortium kombucha (*Zygosaccharomyces* sp., *Bruxellensis* sp. and *Novacetimonas* sp.) [65]. However, monocultures of *Zygosaccharomyces sp* induced the synthesis of specific VOCs such as benzaldehyde, isoamyl alcohol, and phenethyl acetate [65].

A PCA was performed to analyze the variation among the different VOC profile stages (Figure 5). The first dimension explains 68.1% of the total variance, represented on the horizontal axis. A variables correlation plot illustrates the formation of three groups depending on their position on 1:2 plan (Figure 5A). Indeed, green, and black teas are positively correlated together, inducing similar VOC profiles. Then, D0 to D4 fermentation stages are grouped (first quadrant), separated from stages traducing the end of fermentation (D7 to D14) in the fourth quadrant. Moreover, trends provided by the second dimension brought no additional information considering this dataset (explanation of 20.2% of the total variability). To resume, a major significant change in VOC profiles between the beginning (D0-D4) and end (D7-D14) of fermentation stages is highlighted by the PCA. As illustrated by the individual’s representation (Figure 5B), classes that allows the major variability explanation are carboxylic acids, alcohols, and terpenes (most represented along Dim1).

#### 3.2.2. Aromatic Characterization of Kombucha and Establishment of Sensory Profiles

As far as kombucha microorganisms play an important role in the formation of flavor quality during fermentation [30,34], the beverage will now be analyzed from this point of view.

The evolution of most representative aromatic molecules is presented in Table 2. Based on VOC concentrations and perception thresholds in water reported in the literature, odor-activity values were determined along the fermentation process. Odor-activity value (OAV) explains the contribution to a specific aroma. When the OAV is greater than 1, it indicates that the considered compound has a significant aroma contribution in the relative food system. Indeed, the greater the OAV is, the higher the aroma contribution is [1].

The aromatic analysis accounts for 17 VOCs: 5 carboxylic acids (A), 4 alcohols (B), 1 aldehyde (C), 1 ester (D), 1 phenol (E), and 5 terpenes (F). The introduction of a code (letter followed by a number) allows a clear identification of each molecule considering its class, translated afterwards on the sensory profiles (Figure 6).

First, the main VOCs contributing to tea aroma are linalool (OAV of 22.29 in black tea), geraniol (14.68 in black tea), carboxylic acids such as hexadecanoic (7.58 and 5.70 in black and green teas, respectively), nonanoic (1.11 and 1.67 in black and green teas, respectively), and tetradecanoic (3.84 and 3.06 in black and green teas, respectively) acids, to end with methyl 2-hydroxybenzoate (1.70 in black tea) and limonene (2.01 and 4.03 in black and green teas, respectively). Hence, raw matrix presents creamy-waxy aromas from carboxylic acids, with a predominance of floral-sweet and citrus notes (geraniol and linalool) for black tea and citrus-orange (limonene) for green tea. Major tea aromas are chemically formed during the infusion, and bound to precursor handles, such as sugars (carbonaceous substrates available from yeast activity). Moreover, matrix terpenes derivatives contribute to the overall tea aroma because of their oxygenation in the fermentation medium (lower perception threshold inducing a higher OAV) [59].

Along the fermentation process, tea sensory profile is modified through MO activity, resulting in the development of diverse aromatic molecules with different perception profiles. In fact, 2-phenylethanol (OAV of 1.77 at D2 to 13.66 at D14), α-farnesene (from 10.09 at D2 to 5.06 at D14) and p-cymene (the highest OAV of 1.81 at D2) synthesis occur along the fermentation. However, as initial tea constituents globally decrease along fermentation stages, they still contribute to the aroma system as their OAV stays greater than 1 (except for limonene). These changes in the evolution of compounds concentration leads to flavor variations during the fermentation stages (Figure 6): from citrus-floral-sweet, and orange notes at the initial stage (geraniol and linalool domination), stages D2-D4 fermentation brings intense citrus-herbal-lavender-bergamot notes (synthesis of α-farnesene). From D7, while α-farnesene OAV decreases, sweet-floral-bready-honey notes dominates the kombucha flavor (2-phenylethanol synthesis). Moreover, final kombucha flavor (D14) may also present fermented-pungent notes relative to the 3-methylbutan-1-ol OAV increase (OAV of 2.27). Natural plant volatiles such as aliphatic esters, alcohols, acids, and carbonyls are commonly converted into esters, ketones, and alcohols (fatty acids metabolism) during the fermentation process. Oxidation through enzymes, such as lipoxygenase, converts fatty acids to diverse volatile products, depending on MO’s enzymes, pH, and medium conditions [59].

The correlation between changes in the VOC profiles during the fermentation process and metabolic pathways is reported in the literature. Therefore, aroma-active compounds also fluctuate [1,34,66]. Considering the metabolic pathways employed in the kombucha fermentation process, multiple aromatic compounds are referred to be microbial metabolites (2-phenyl alcohol [62]) [36]. Terpene synthesis by diverse yeast taxa at the beginning of fermentation is again highlighted here [62], through the highest aroma perception of α-farnesene and p-cymene at D4 and D2, respectively. However, although the final flavor of kombucha maintains characteristic aroma compounds from its original substrate, it may have several common compounds (such as 2-phenyl ethanol and nonanoic acid) that contributes to the signature aroma of kombucha [36]. The final flavor of kombucha also depends on fermentation parameters such as the initial sucrose concentration and kombucha starter, raw matrix, microbial SCOBY composition, temperature, and time [7,17,19,35,67].

Eventually, through the aroma-active compound analysis, overall beverage flavor and taste are determined. Despite the need of sensory analysis to evaluate consumer acceptance, the fermentation time control seems to give an overview of potential developed aromas and perceptions. Indeed, the obtention of fruity-, floral-, acidic-, and citrus-like beverages could be intended by the fermentation control, not to mention the primary safety control of these fermented beverages (food safety conditions avoiding any pathogen development).

To go deeper in the aroma-active compound analysis, an insight concerning the temperature impact on the beverage flavor is studied. The consideration of an extra fermentation temperature in the study (20 °C) allows the qualitative comparison of the development of aroma active compounds. Based on the following sensory profiles, flavor comparison is illustrated for the intermediate (D7) and final (D14) stages (Figure 7A,B respectively). Considering both fermentation stages, the temperature seems to have a positive correlation on the alcohol synthesis (except for an observed inhibition of B1 synthesis at D7-30 °C). Otherwise, temperature may inhibit the synthesis of carboxylic acids and phenols (D14). The underlying hypothesis arising from these observations is the role of temperature as a factor in the metabolite production rate in the fermentation media. As the production of aroma-active compounds obtained at 30 °C-D14 is not like that obtained at 20 °C-D7, the relationship between metabolite production and temperature might be more complex. In fact, as most aromatic compounds developed during fermentation are synthetized through metabolic pathways, the MOs involved in the process might have optimal temperature range conditions [68]. For instance, yeast taxa allowing the synthesis of multiple aroma-active compounds prefer fermentation temperatures below 30 °C [68].

Finally considering the final flavor approach, a higher temperature could inhibit the formation of certain aroma compound classes such as carboxylic acids and phenols (respectively responsible for sour-creamy-fatty and spicy-woody notes). On the contrary, alcohol synthesis might be improved with temperature, accounting for a greater perception of floral-sweet-citrus-like flavors. This information allows us to consider the temperature as another way of controlling fermentation flavors, not to mention that major fermentation modalities may be linked with each other (fermentation time for instance). The effect of crossed fermentation parameters (temperature and time) on the flavor development should be further studied, towards the control of fermentation and the production of quality flavored beverages.

## 4. Conclusions

In the present study, the main kinetics including pH, major fermentation components, VOCs, and aroma-active compounds were analyzed during kombucha fermentation. Concerning the physico-chemical characterization of kombucha, a pH decrease reflects an acidification in the fermentation medium and illustrates organic acid synthesis by the fermentation microbiome. In these conditions, low pH (below 4.5) ensures food safety by inhibiting spoilage and pathogen development.

Through the kinetics of major compounds, the main microbial consortium pathways are highlighted, such as glycolysis pathway and AAB pathways.

Dynamic changes in VOCs were investigated using SBSE-GC-MS during kombucha fermentation. A total of 87 VOCs were detected in kombucha during the fermentation stages, including 28 carboxylic acids, 23 alcohols, 10 ketones, 9 terpenes, 7 esters, 7 aldehydes, 2 phenols and 1 benzene. The fermentation process changes the VOC tea profile dynamically into a different mixture of compounds. The synthesis of mainly phenethyl alcohol and isoamyl alcohol probably by *Saccharomyces* genus leads to ester formation (ethyl acetate main structure). Moreover, the terpene synthesis occurring at the beginning of fermentation (Δ-3-carene, α-phellandrene, γ-terpinene, and m- and p-cymene) could be related to yeast activity as well.

The aromatic analysis, based on odor-active values, accounts for 17 aroma-active compounds. These changes in the evolution of compound concentrations led to flavor variations at the different fermentation stages: from citrus-floral-sweet and orange notes at the initial stage (geraniol and linalool domination), with stages D2-D4 fermentation bringing intense citrus-herbal-lavender-bergamot notes (synthesis of α-farnesene). From stage D7, sweet-floral-bready-honey notes dominates the kombucha flavor (2-phenylethanol synthesis). However, although the final flavor of kombucha keeps characteristic aroma compounds from its original substrate, several common compounds (such as 2-phenyl ethanol, nonanoic, and octanoic acids) may also contribute to the signature aroma of kombucha.

## 5. Prospects of this Research Study

Firstly, the characterization of the microbial beverage population could confirm some existing relations between VOC synthesis and metabolic pathways involved during fermentation, not only by identifying the microorganisms present in the starting SCOBY but also by studying its evolution and potential specification. Secondly, despite the need for sensory analysis to evaluate consumer acceptance and preferences considering the beverage, the fermentation time control seems to give an overview of potentially developed aromas and perceptions. Moreover, as temperature may have an impact on the final kombucha flavor, the effect of crossed fermentation parameters (temperature and time) on the development of aroma-active compounds should be further studied, towards the control of fermentation and the production of quality flavored beverages.

As far as health beverage aspects are concerned, it would be interesting to investigate the potential antioxidant and anti-inflammatory properties. Indeed, the evolution of these parameters with fermentation modalities could help developing optimal functional beverages. Moreover, interactions between the beverage and the intestinal consumer microbiomes should be further studied.

In this context, the use of alternative fermentation matrices could be intended. Several studies [14,17,26,69,70,71] demonstrated the efficiency of mixed fermentation on various matrices such as tea, hibiscus, coconut water, cereals, etc. A panel of functional beverages of this type could help in determining the limit of the necessary nutrients for the SCOBY development. The study of kinetics of the tea constituents may allow the evaluation of this fermentation substrate impact on health. The use of lactose and gluten-free matrices could aim at satisfying a larger target population. Indeed, at the dawn of a food transition encouraging the consumption of healthy and sustainable functional foods, it is now crucial to promote food with health benefits for the consumer.

## Figures and Tables

**Figure 1 foods-12-01657-f001:**
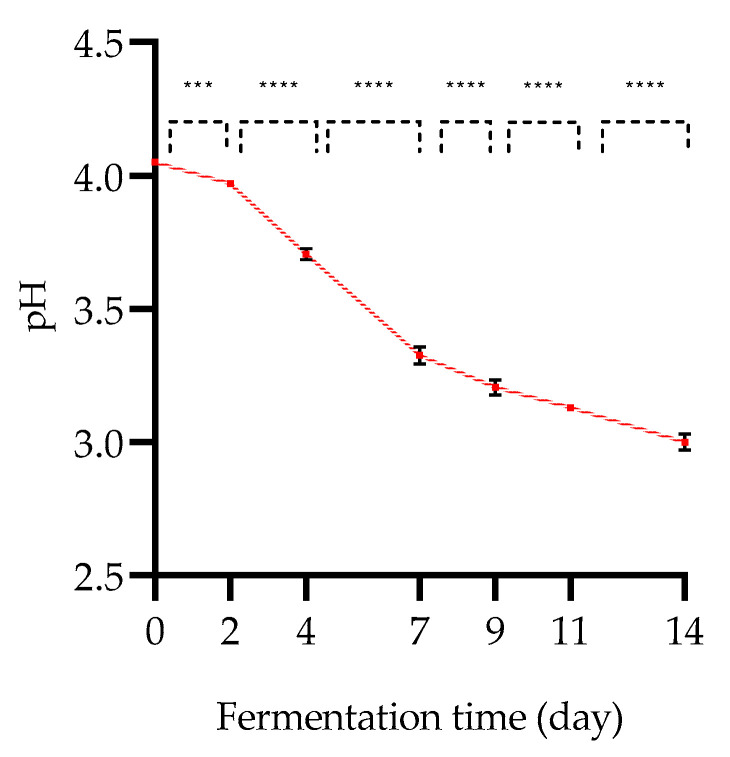
pH evolution of kombucha samples as a function of the fermentation time (day) at 30 °C (mean ± SD; n = 3; Student’s *t*-test; ****, *p* < 0.0001; ***)).

**Figure 2 foods-12-01657-f002:**
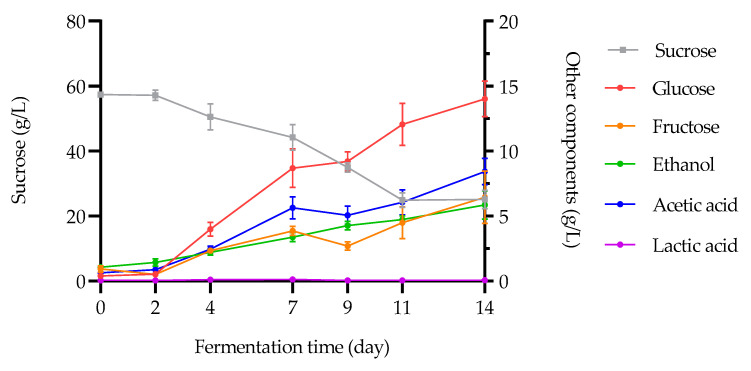
Evolution of the concentration of carbohydrates, alcohol, and acids (g/L) as functions of fermentation time (day) at 30 °C (mean ± SD; n = 3).

**Figure 3 foods-12-01657-f003:**
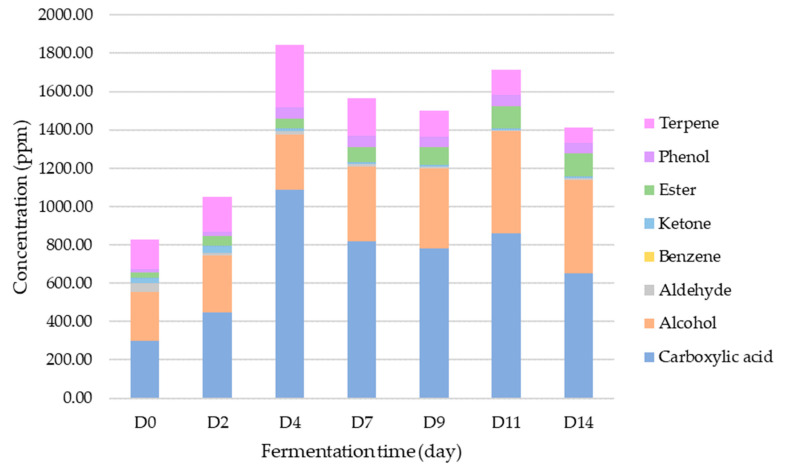
Changes in average contents (ppm) of different classes of VOCs with kombucha fermentation time (day) at 30 °C.

**Figure 4 foods-12-01657-f004:**
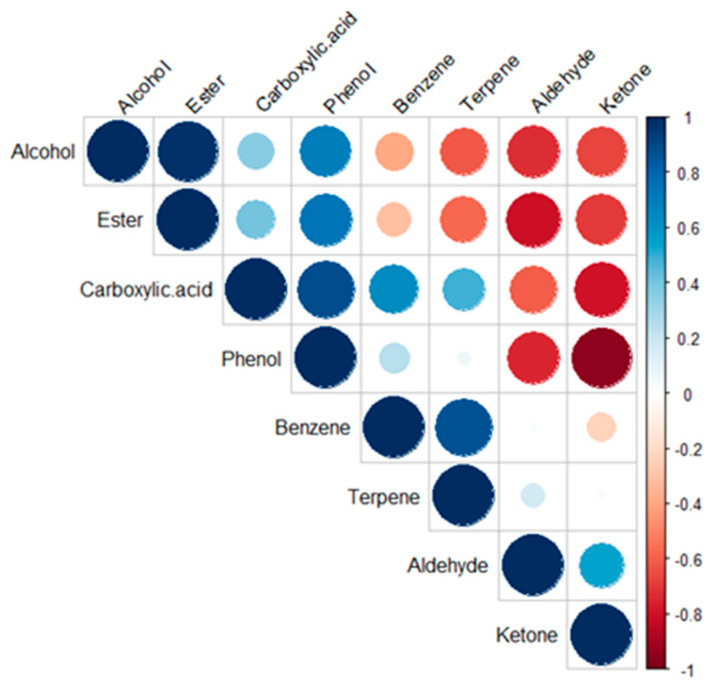
Correlation between different classes of VOCs during kombucha fermentation stages conducted at 30 °C (a color gradient denotes the Spearman’s correlation coefficients).

**Figure 5 foods-12-01657-f005:**
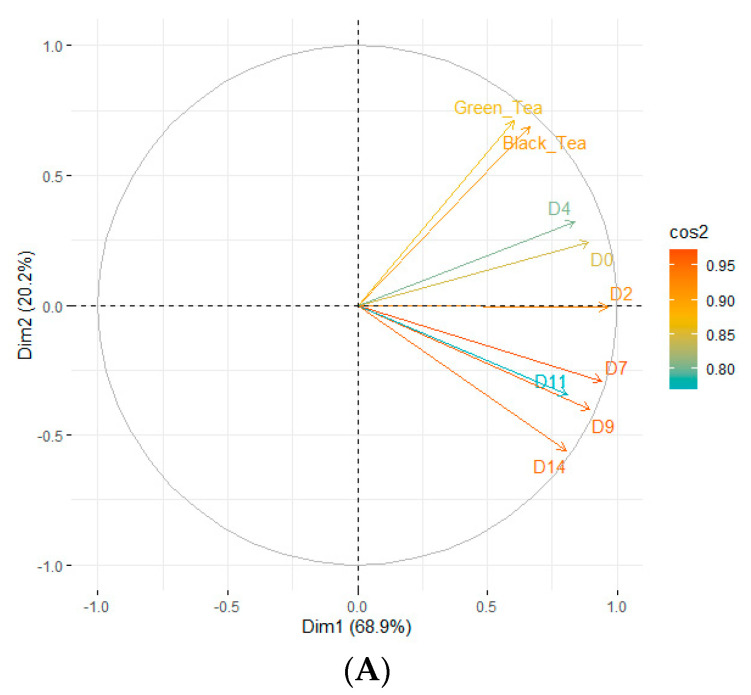
Principal component analysis (mean values for VOCs averaged across kombucha fermentation time (day)) plots of variables correlation (cos² color scale) (**A**) and individuals’ representation (ellipses corresponds to VOCs classes) (**B**) (labelled individuals are those best represented on the plan).

**Figure 6 foods-12-01657-f006:**
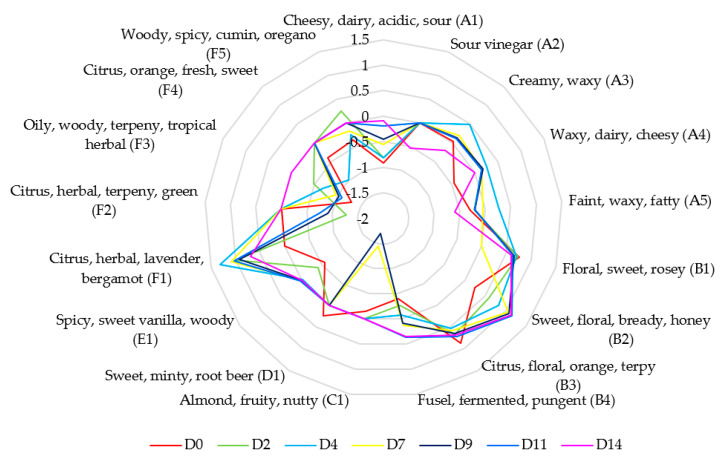
Estimated sensory profile is expressed as the log of the OAV for most representative volatile organic compounds (%area > 1%) with kombucha fermentation time (day) at 30 °C (OAV = concentration/perception threshold).

**Figure 7 foods-12-01657-f007:**
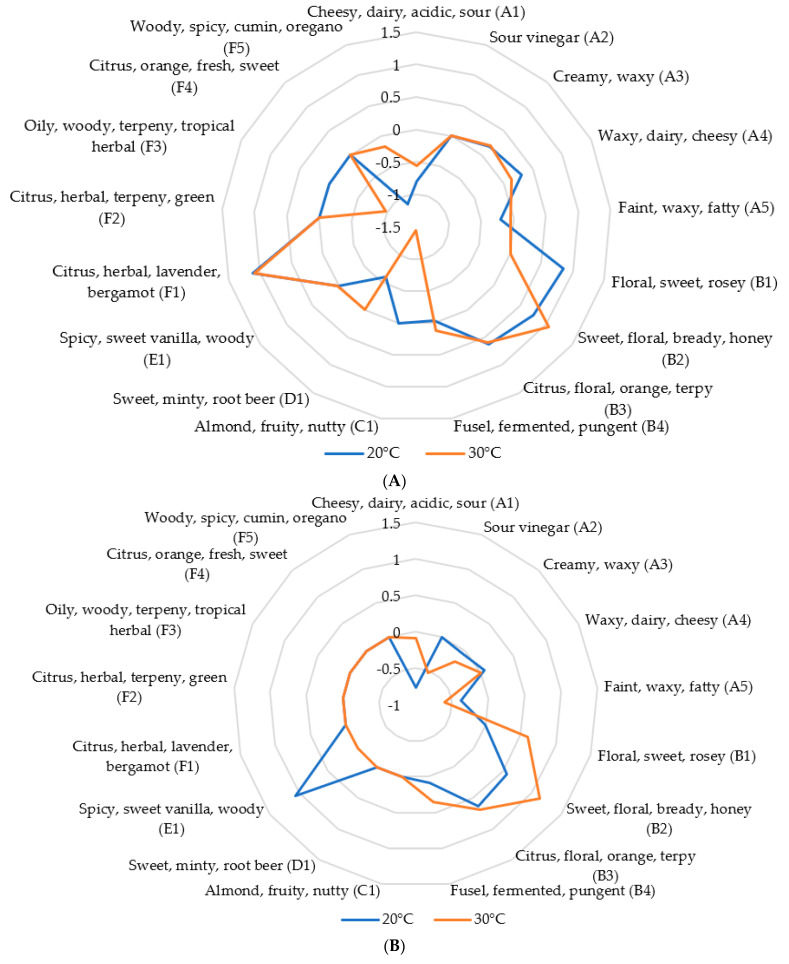
Variation of the estimated sensory profile (expressed as the log of OAV; OAV = concentration/perception threshold) of the most representative volatile compounds (%area > 1%) with temperature (°C) on kombucha fermentation in the (**A**) intermediate stage (D7) and (**B**) the final stage (D14).

**Table 1 foods-12-01657-t001:** Volatile organic compounds contents ^1^ (ppm) with kombucha fermentation time (day) conducted at 30 °C.

					Raw Matrix		Kombucha Fermentation Stages						
Class	Compound	CAS Number	RI ^2^	RI *lit* ^3^	Black Tea	Green Tea	D0	D2	D4	D7	D9	D11	D14
Carboxylic acid	(2E)-3,7-dimethylocta-2,6-dienoic acid	4698-08-2	2326	2329	nd	nd	5.02 ± 0.26	4.68 ± 0.84	4.86 ± 0.34	5.24 ± 0.78	6.40 ± 0.91	7.08 ± 0.81	5.93 ± 1.47
	(9E,12E)-octadeca-9,12-dienoic acid	506-21-8	3171	3168	nd	nd	nd	nd	7.15 ± 0.50	nd	nd	nd	nd
	(E)-hexadec-9-enoic acid	10030-73-6	2570	2579	226.30	129.69	nd	nd	2.36 ± 0.53	nd	7.81 ± 4.21	10.96 ± 1.02	nd
	(E)-oct-2-enoic acid	1871-67-6	2084	2082	nd	nd	1.84 ± 0.66	1.26 ± 0.29	nd	nd	nd	nd	nd
	(Z)-heptadec-10-enoic acid	29743-97-3	3036	3030	30.22	nd	8.99 ± 0.15	2.09 ± 0.66	19.46 ± 6.74	7.55 ± 1.18	4.45 ± 3.00	7.35 ± 0.46	2.18 ± 0.63
	(Z)-hexadec-7-enoic acid	2416-19-5	2920	2924	nd	nd	nd	nd	14.03 ± 16.43	nd	33.57 ± 8.63	nd	nd
	(Z)-hexadec-9-enoic acid	373-49-9	2932	2944	nd	nd	21.21 ± 0.84	45.90 ± 3.93	123.34 ± 2.55	58.62 ± 5.45	41.50 ± 13.58	57.22 ± 11.48	17.34 ± 10.54
	(Z)-nonadec-10-enoic acid	73033-09-7	3258	3256	nd	nd	nd	29.08 ± 6.69	nd	28.66 ± 2.64	nd	nd	nd
	(Z)-octadec-11-enoic acid	506-17-2	3132	3135	nd	nd	nd	nd	nd	37.21 ± 12.61	26.58 ± 6.89	33.99 ± 7.19	9.91 ± 3.30
	(Z)-octadec-9-enoic acid	112-80-1	3133	3142	nd	198.03	nd	nd	71.75 ± 98.09	nd	nd	nd	nd
	(Z)-tetradec-9-enoic acid	544-64-9	3168	3173	19.90	nd	nd	1.82 ± 0.21	7.57 ± 6.16	2.59 ± 1.01	2.62 ± 1.34	2.92 ± 1.98	nd
	2-methylpropanoic acid	79-31-2	1565	1563	nd	nd	nd	nd	2.71 ± 0.04	3.03 ± 0.30	2.93 ± 1.08	5.84 ± 0.40	8.13 ± 1.48
	3-methylbutanoic acid	503-74-2	1665	1664	nd	nd	5.81 ± 0.47	7.66 ± 1.41	7.74 ± 0.21	13.86 ± 0.88	17.54 ± 4.19	31.95 ± 3.24	40.74 ± 10.26
	acetic acid	64-19-7	1450	1450	nd	nd	nd	nd	nd	nd	nd	nd	15.17 ± 9.76
	butanoic acid	107-92-6	1623	1628	nd	nd	nd	nd	4.15 ± 1.73	1.86 ± 0.23	nd	nd	nd
	dec-9-enoic acid	14436-32-9	2300	2305	nd	nd	nd	nd	nd	nd	nd	nd	0.99 ± 0.24
	decanoic acid	334-48-5	2264	2267	21.83	17.10	13.13 ± 1.01	3.48 ± 0.27	30.58 ± 0.74	62.49 ± 1.74	71.98 ± 4.28	94.60 ± 3.52	89.08 ± 7.94
	dodecanoic acid	143-07-7	2474	2471	11.96	7.04	3.03 ± 0.67	1.84 ± 0.48	11.33 ± 3.14	4.83 ± 0.60	3.71 ± 1.03	4.72 ± 0.84	1.98 ± 0.61
	heptadecanoic acid	506-12-7	3027	3027	19.93	nd	nd	4.74 ± 0.96	14.57 ± 4.82	6.35 ± 1.17	5.02 ± 0.35	5.46 ± 1.72	2.15 ± 0.28
	heptanoic acid	111-14-8	1944	1946	2.20	nd	7.97 ± 0.89	6.44 ± 1.53	6.65 ± 2.39	7.24 ± 1.91	9.32 ± 1.77	9.64 ± 1.36	9.07 ± 1.79
	hexadecanoic acid	57-10-3	2898	2890	568.52	427.35	80.04 ± 0.21	97.89 ± 8.45	234.64 ± 41.41	117.93 ± 11.21	100.84 ± 28.58	98.64 ± 16.32	47.75 ± 12.37
	hexanoic acid	142-62-1	1838	1838	nd	nd	21.23 ± 0.92	29.42 ± 6.91	43.10 ± 3.45	56.00 ± 9.57	44.22 ± 6.62	60.84 ± 10.69	51.37 ± 9.61
	nonanoic acid	112-05-0	2171	2174	11.07	nd	16.73 ± 0.07	3.54 ± 0.79	14.45 ± 0.22	18.17 ± 0.18	13.72 ± 0.61	14.84 ± 1.65	13.91 ± 1.11
	octadecanoic acid	57-11-4	3107	3104	11.36	nd	5.01 ± 0.60	8.66 ± 3.73	37.79 ± 6.50	12.45 ± 1.02	32.22 ± 1.67	22.59 ± 2.55	13.44 ± 8.88
	octanoic acid	124-07-2	2052	2050	5.44	6.07	74.71 ± 0.69	164.58 ± 15.00	284.03 ± 15.47	313.86 ± 10.05	321.71 ± 17.22	339.21 ± 23.23	303.54 ± 9.94
	pentadecanoic acid	1002-84-2	2741	2745	150.78	102.42	16.04 ± 0.65	24.28 ± 3.39	74.71 ± 23.23	36.10 ± 5.89	23.62 ± 8.84	30.41 ± 7.96	10.27 ± 2.85
	tetradecanoic acid	544-63-8	2656	2660	134.58	107.04	18.04 ± 0.08	21.12 ± 1.67	65.28 ± 17.97	31.26 ± 3.45	22.05 ± 11.59	22.07 ± 3.98	8.83 ± 2.93
	tridecanoic acid	638-53-9	2580	2573	8.57	nd	nd	nd	4.12 ± 1.12	1.38 ± 0.04	1.04 ± 0.45	1.22 ± 0.07	nd
	Total				1222.65	994.75	298.2	448.38	1086.38	816.65	781.66	861.16	653.95
Alcohol	geraniol	106-24-1	1840	1836	44.04	nd	18.03 ± 0.79	16.34 ± 0.71	15.32 ± 0.80	nd	13.88 ± 0.91	12.78 ± 1.80	12.09 ± 1.74
	nerol	106-25-2	1793	1794	3.28	nd	1.98 ± 0.13	2.14 ± 0.09	2.11 ± 0.18	2.70 ± 0.07	3.08 ± 0.19	3.40 ± 0.19	3.16 ± 0.11
	linaloolpyran oxide C	10448-31-4	1755	1741	9.58	nd	1.91 ± 0.11	3.10 ± 0.55	1.87 ± 0.21	1.53 ± 0.12	1.28 ± 0.25	1.67 ± 0.29	1.32 ± 0.07
	hotrienol	29957-43-5	1608	1613	nd	2.09	nd	1.94 ± 0.52	1.41 ± 0.04	1.19 ± 0.32	1.22 ± 0.27	1.83 ± 0.07	2.09 ± 0.41
	(Z)-hex-3-en-1-ol	928-96-1	1383	1390	nd	nd	1.11 ± 0.40	2.34 ± 0.81	2.10 ± 0.06	1.83 ± 0.06	1.77 ± 0.33	2.13 ± 0.33	1.79 ± 0.18
	1-methoxypropan-2-ol	107-98-2	1152	1160	nd	nd	47.78 ± 1.10	47.04 ± 18.20	9.54 ± 5.04	9.64 ± 1.81	17.69 ± 5.07	17.89 ± 3.73	nd
	8-camphenemethanol	2226-05-3	1688	1682	nd	nd	1.11 ± 0.15	0.92 ± 0.07	nd	nd	nd	nd	nd
	linalool oxide	60047-17-8	1440	1442	nd	nd	10.87 ± 1.01	10.03 ± 1.01	nd	nd	nd	nd	nd
	2,6-dimethylcyclohexan-1-ol	5337-72-4	1112	1114	nd	nd	5.53 ± 0.70	4.27 ± 0.43	2.83 ± 0.06	2.08 ± 0.01	1.38 ± 0.42	1.75 ± 0.10	1.38 ± 0.00
	(-)-α-terpineol	10482-56-1	1690	1690	nd	nd	nd	13.22 ± 0.76	11.34 ± 1.32	13.22 ± 0.35	13.16 ± 1.05	18.11 ± 1.45	16.52 ± 1.24
	trans-furan linalool oxide	34995-77-2	1468	1460	109.31	nd	15.95 ± 0.35	15.90 ± 1.93	10.64 ± 0.30	7.97 ± 0.27	5.95 ± 1.08	6.95 ± 1.02	5.28 ± 0.37
	2-ethylhexan-1-ol	104-76-7	1489	1490	10.06	6.86	3.95 ± 0.10	8.66 ± 0.60	9.18 ± 0.41	11.76 ± 0.42	7.75 ± 0.39	7.83 ± 1.38	7.94 ± 0.28
	2-phenylethanol	60-12-8	1897	1890	1.97	nd	35.83 ± 0.53	77.78 ± 13.55	135.58 ± 3.04	225.04 ± 5.48	242.05 ± 20.25	288.59 ± 26.39	273.15 ± 2.16
	linalool	78-70-6	1546	1545	222.28	5.44	72.18 ± 2.10	45.82 ± 2.63	33.83 ± 1.99	38.39 ± 3.18	43.87 ± 4.91	53.14 ± 6.28	48.69 ± 2.55
	3-methylbutan-1-ol	123-51-3	1216	1218	nd	nd	22.30 ± 3.55	27.37 ± 8.73	41.89 ± 0.86	66.49 ± 11.82	61.35 ± 29.29	115.60 ± 28.99	113.39 ± 8.02
	4,8-dimethylnona-1,7-dien-4-ol	17920-92-2	1182	1191	nd	nd	nd	1.27 ± 0.17	nd	nd	nd	nd	nd
	4-carvomenthenol	562-74-3	1597	1594	1.34	nd	1.29 ± 0.42	0.88 ± 0.12	nd	nd	nd	nd	nd
	butan-1-ol	71-36-3	1157	1157	nd	1.01	nd	3.29 ± 2.61	3.18 ± 1.57	2.96 ± 1.24	nd	3.12 ± 2.19	2.77 ± 0.16
	heptan-3-ol	589-82-2	1317	1306	nd	nd	nd	1.50 ± 0.39	nd	nd	nd	nd	nd
	hexan-1-ol	111-27-3	1354	1356	7.27	nd	10.88 ± 0.53	5.23 ± 1.02	1.75 ± 0.23	1.27 ± 0.14	nd	nd	nd
	nonan-1-ol	143-08-8	1657	1666	1.12	nd	1.89 ± 0.17	2.38 ± 0.13	3.14 ± 0.03	1.62 ± 0.21	nd	nd	nd
	oct-1-en-3-ol	3391-86-4	1450	1456	3.80	nd	1.91 ± 0.09	1.00 ± 0.24	nd	nd	nd	nd	nd
	octan-1-ol	111-87-5	1556	1565	nd	nd	5.44 ± 0.77	2.73 ± 0.62	4.30 ± 0.46	3.33 ± 0.31	1.76 ± 0.19	nd	nd
	Total				414.05	15.39	260.25	294.48	290.03	390.63	415.71	528.83	484.51
Aldehyde	β-cyclocitral	432-25-7	1613	1611	13.34	30.77	5.63 ± 0.56	1.96 ± 0.11	nd	nd	nd	nd	nd
	2-phenylacetaldehyde	122-78-1	1635	1632	nd	nd	nd	nd	nd	nd	nd	nd	1.35 ± 0.00
	3-methylbenzaldehyde	620-23-5	1638	1635	nd	nd	nd	nd	3.19 ± 0.04	nd	nd	nd	nd
	benzaldehyde	100-52-7	1611	1617	10.54	nd	34.41 ± 1.19	nd	nd	1.78 ± 0.05	1.02 ± 0.05	nd	nd
	decanal	112-31-2	1499	1498	10.80	nd	nd	2.20 ± 0.36	2.65 ± 0.34	2.22 ± 0.2	1.61 ± 0.21	nd	1.34 ± 0.05
	hexanal	66-25-1	1099	1094	10.02	22.54	9.07 ± 0.69	6.32 ± 0.44	4.73 ± 0.43	2.96 ± 1.75	1.93 ± 0.40	2.68 ± 0.42	nd
	nonanal	124-19-6	1395	1390	34.69	nd	nd	7.06 ± 1.00	7.07 ± 0.08	6.57 ± 1.11	4.94 ± 0.43	4.86 ± 1.49	4.13 ± 0.39
	Total				79.40	53.32	49.12	14.71	17.63	13.54	9.51	7.53	6.37
Benzene	styrene	100-42-5	1258	1259	17.44	62.63	nd	1.60 ± 0.98	nd	nd	nd	nd	nd
	Total				17.44	62.63	nd	1.60	nd	nd	nd	nd	nd
Ketone	cis-dihydrocarvone	3792-53-8	1600	1602	nd	nd	nd	2.70 ± 0.19	nd	nd	nd	nd	nd
	(3E)-6-methylhepta-3,5-dien-2-one	1604-28-0	1588	1582	nd	nd	1.81 ± 0.32	1.14 ± 0.30	nd	nd	nd	nd	nd
	(3E,5E)-octa-3,5-dien-2-one	30086-02-3	1566	1565	nd	nd	5.08 ± 0.06	14.85 ± 1.51	nd	nd	nd	nd	nd
	dihydroactinidiolide	17092-92-1	2291	2291	7.90	96.50	7.95 ± 0.16	7.98 ± 0.97	6.98 ± 0.37	5.80 ± 0.42	5.37 ± 0.90	6.35 ± 0.70	5.90 ± 0.31
	dehydro-β-ionone	1203-08-3	1927	1932	nd	nd	nd	nd	nd	1.06 ± 0.07	nd	1.34 ± 0.19	1.30 ± 0.24
	β-ionone	14901-07-6	1964	1967	6.15	23.29	2.77 ± 0.17	1.63 ± 0.06	nd	nd	nd	nd	nd
	1-(5-methylthiophen-2-yl)ethanone	13679-74-8	1185	1197	0.89	6.45	nd	nd	nd	nd	2.13±	1.27 ± 0.25	1.24 ± 0.12
	β-ionone epoxyde	23267-57-4	1963	1968	6.56	61.74	4.26 ± 0.40	3.52 ± 0.16	2.89 ± 0.04	2.06 ± 0.12	1.44 ± 0.15	nd	nd
	5-methylheptan-2-one	18217-12-4	1145	1152	4.58	91.06	nd	nd	nd	nd	nd	nd	2.26 ± 0.34
	6-methylhept-5-en-2-one	110-93-0	1337	1340	12.82	3.09	7.01 ± 0.29	3.94 ± 0.75	2.04 ± 0.14	2.19 ± 0.47	2.08 ± 0.05	2.17 ± 0.20	1.38 ± 0.31
	Total				38.90	282.12	28.87	35.77	11.91	11.12	11.69	10.41	11.33
	(1-ethynylcyclohexyl) carbamate	126-52-3	1316	1313	nd	nd	nd	nd	3.37 ± 0.03	nd	nd	nd	nd
	2-(1H-indol-3-yl)ethyl acetate	13137-14-9	1863	1865	nd	nd	nd	nd	10.21 ± 4.16	20.53 ± 3.19	31.63 ± 6.01	25.18 ± 1.11	20.59 ± 1.53
	2-phenylethyl acetate	103-45-7	1806	1808	nd	nd	4.63 ± 0.53	3.93 ± 0.56	7.12 ± 0.53	12.93 ± 1.03	19.63 ± 3.23	28.51 ± 2.47	40.66 ± 4.15
Ester	ethyl 2-phenylacetate	101-97-3	1777	1763	nd	nd	nd	nd	nd	nd	1.25 ± 0.13	1.55 ± 0.07	1.86 ± 0.06
	ethyl hexanoate	123-66-0	1239	1241	nd	nd	nd	nd	nd	2.36 ± 0.38	3.31 ± 0.76	3.99 ± 1.13	5.87 ± 1.53
	methyl 2-hydroxybenzoate	119-36-8	1764	1766	17.03	nd	17.61 ± 0.59	10.05 ± 0.86	nd	nd	nd	nd	nd
	methyl hexanoate	106-70-7	1190	1184	nd	nd	1.45 ± 0.67	3.60 ± 3.41	2.50 ± 0.73	nd	1.46 ± 0.28	2.73 ± 0.52	2.73 ± 0.24
	Total				17.03	0.00	23.70	16.38	15.47	35.81	46.74	61.44	71.71
Phenol	4-ethyl-2-methoxyphenol	2785-89-9	2016	2020	nd	nd	7.86 ± 0.39	11.82 ± 1.11	29.02 ± 1.53	30.88±	30.85±	32.44 ± 2.46	29.59 ± 1.71
	4-ethylphenol	123-07-9	2161	2164	nd	nd	7.99 ± 0.08	10.54 ± 1.18	28.53 ± 0.10	29.04±	27.10±	28.74 ± 3.23	24.46 ± 2.22
	Total				nd	nd	16.02	22.36	57.55	59.92	57.95	61.18	54.06
	α-farnesene	502-61-4	1687	1695	nd	nd	nd	nd	nd	nd	nd	nd	1.078 ± 0.02
	(-)-limonene	5989-54-8	1027	1027	nd	nd	12.89 ± 0.98	11.87 ± 15.94	nd	nd	nd	nd	nd
	squalene	111-02-4	2865	2865	563.30	510.25	139.59 ± 0.65	151.29 ± 17.3	311.75 ± 73.96	184.91 ± 27.17	125.47 ± 9.83	158.35 ± 10.83	75.91 ± 20.61
	m-cymene	535-77-3	1272	1270	10.39	33.08	nd	1.06 ± 0.30	nd	nd	2.44 ± 0.09	nd	nd
Terpene	limonene	138-86-3	1194	1195	60.28	121.01	nd	13.01 ± 6.44	8.28 ± 0.50	4.28 ± 1.10	3.69 ± 1.06	3.41 ± 0.57	nd
	p-cymene	99-87-6	1266	1268	nd	nd	2.74 ± 0.49	nd	nd	nd	nd	3.23 ± 0.91	nd
	γ-terpinene	99-85-4	1242	1242	1.90	5.59	1.43 ± 0.77	1.70 ± 0.23	nd	nd	nd	nd	nd
	α-phellandrene	99-83-2	1167	1166	nd	nd	nd	5.36 ± 2.48	2.70 ± 1.32	2.09 ± 0.78	2.05 ± 0.23	nd	nd
	Δ-3-carene	498-15-7	1152	1150	nd	nd	nd	9.96 ± 1.60	nd	3.68 ± 1.31	nd	4.91 ± 2.67	2.73 ± 0.47
	Total				635.88	669.93	156.66	183.89	322.73	194.97	131.60	168.77	78.45

1. Data are expressed as the mean ± SD; “nd”: Not detected. 2. RI: Retention indices on Carbowax 20M column were determined by n-alkanes. 3. RI *lit*: Literature Retention indices reported for polar capillary Carbowax 20M column [45].

**Table 2 foods-12-01657-t002:** OAV evolution of most representative volatile organic compounds (%area > 1%) with kombucha fermentation time (day) conducted at 30 °C (OAV = concentration/perception threshold).

								Raw Matrix		Kombucha Fermentation Stages						
Class		Compound	CAS	RI ^1^	RI *li* ^2^	Perception Threshold (ppm) [46]	Aroma Description [46]	Black Tea	Green Tea	D0	D2	D4	D7	D9	D11	D14
Carboxylic acid	A1	3-methylbutanoic acid	503-74-2	1665	1664	50	Cheesy, dairy, acidic, sour	nd	nd	0.12	0.15	0.15	0.28	0.35	0.64	0.81
	A2	acetic acid	546-67-8	1450	1450	50.5	Sour vinegar	nd	nd	nd	nd	nd	nd	nd	nd	0.30
	A3	hexadecanoic acid	57-10-3	2898	2890	75	Creamy, waxy	7.58	5.70	1.07	1.30	3.13	1.57	1.34	1.31	0.64
	A4	nonanoic acid	112-05-0	2171	2174	10	Waxy, dairy, cheesy	1.11	1.67	0.35	1.44	1.82	1.37	1.48	1.39	nd
	A5	tetradecanoic acid	544-63-8	2656	2660	35	Faint, waxy, fatty	3.84	3.06	0.51	0.60	1.86	0,89	0.63	0.63	0.25
Alcohol	B1	geraniol	106-24-1	1840	1836	3	Floral, sweet, rosy	14.68	nd	5.82	5.45	5.11	nd	4.63	4.26	4.03
	B2	2-phenylethanol	60-12-8	1897	1890	20	Sweet, floral, bready, honey	0.10	nd	1.77	3.89	6.78	11.25	12.10	14.43	13.66
	B3	linalool	78-70-6	1546	1545	10	Citrus, floral, orange, terpeny	22.29	0.54	7.37	4.58	3.38	3.84	4.39	5.31	4.87
	B4	3-methylbutan-1-ol	123-51-3	1216	1218	50	Fusel, fermented, pungent	nd	nd	0.40	0.54	0.84	1.33	1.23	2.31	2.27
Aldehyde	C1	benzaldehyde	100-52-7	1611	1617	50	Almond, fruity, nutty	0.21	nd	0.70	nd	nd	0.04	0.02	nd	nd
Ester	D1	methyl 2-hydroxybenzoate	119-36-8	1764	1766	10	Sweet, minty, root beer	1.70	nd	1.72	1.00	nd	nd	nd	nd	nd
Phenol	E1	4-ethyl-2-methoxyphenol	2785-89-9	2016	2020	30	Spicy, sweet vanilla, woody	nd	nd	0.27	0.39	0.97	1.03	1.03	1.08	0.98
Terpene	F1	α-farnesene	502-61-4	1687	1695	15	Citrus, herbal, lavender, bergamot	nd	nd	nd	10.09	20.78	12.33	8.36	10.56	5.06
	F2	α-phellandrene	99-83-2	1167	1166	20	Citrus, herbal, terpeny, green	nd	nd	nd	0.05	nd	nd	0.12	0.17	nd
	F3	γ-terpinene	99-85-4	1242	1242	40	Oily, woody, terpeny, tropical herbal	0.05	0.14	0.05	0.32	0.21	0.11	0.09	0.08	nd
	F4	limonene	138-86-3	1194	1195	30	Citrus, orange, fresh, sweet	2.01	4.03	0.41	nd	0.10	nd	nd	nd	nd
	F5	p-cymene	99-87-6	1266	1268	5.5	Woody, spicy, cumin, oregano	nd	nd	0.43	1.81	0.56	0.67	nd	nd	nd

1. RI: Retention indices on Carbowax 20M column were determined by n-alkanes. 2. RI *lit*: Literature Retention indices reported for polar capillary Carbowax 20M column [45].

## Data Availability

The datasets generated for this study are available on request to the corresponding author.

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
