# Peer review of "Characterization of Aroma Active Compound Production during Kombucha Fermentation: Towards the Control of Sensory Profiles"

_foods, 2023, doi:10.3390/foods12081657_

Round 1

Reviewer 1 Report

Comments and Suggestions for Authors

Dear Autors,

I am writing to express my opinion regarding the publication entitled "Characterization of aroma active compounds production during kombucha fermentation: towards the control of sensory profiles". Based on my evaluation, I would like to bring to your attention a concern that I have regarding the production of lactic acid (lines 246-249).

Specifically, I am unsure if it is appropriate to refer to the increase in lactic acid concentration from 0.05 g/L to 0.06 g/L during the process as "synthesis" of lactic acid. I wonder if this increase falls within the margin of error for measurement and could possibly be a result of experimental variability.

I would appreciate it if the authors could provide further clarification or additional data to support their claim of lactic acid synthesis during the fermentation process.

Thank you for considering my suggestion.

Author Response

Dear Reviewer,

I would like to thank you for the proofreading of my article and the relevant comments and remarks established, contributing to the improvement of my work.

Please find enclosed my response relative to the Lactic acid synthesis concern you mentionned in this report. 

Hoping that this explanation would meet your expectations.

Best regards,

Suffys Sarah

Reviewer 2 Report

Comments and Suggestions for Authors

The manuscrpit is of high interest. However, materials and methods information are missing, description of methodology needs revision. Major concern reffers to the identification of VOCs and how the OAV studies were conducted. A experiment was conducted using a different temperature, however the results were only presented for OAV.

On the manuscript attached some modifications are suggested.

Author Response

Dear Reviewer,

I would like to thank you for the proofreading of my article and the relevant comments and remarks established, contributing to the improvement of my work.

Please find attached to this letter the corrected version of the manuscript, based on your comments and remarks, highlighted with the modification marks mode.

  1. Concerns regarding the Material & Methods 2.4 Chromatographic analysis of the development of volatile organic compounds manuscript section
    1. Match factor ≥ 70%

Response:

Thank you for your pertinent comment aimed at improving my work. Indeed, I agree that the set value of the match factor seems relatively low. Please find below an explanation of the approach to this issue.

Regarding the volatile organic compound’s identification and data processing, filters were applied to raw data. The match factor, meaning the percentage of probability of correct identification of the compound under consideration, was implemented ≥ 70%. This first rough step of molecule selection by the software allowed me to perform a first sorting on the data set obtained.

Despite the increase in uncertainty concerning the identification of the compound considered by the software (Mass Hunter Analysis), a crucial following step in the identification of the molecules was performed manually for each molecule. This disciplined treatment approach consisted in the validation of all molecules one by one, referring to the mass spectrum by consulting the spectra of reference molecules in the literature. Additionally, RI identification (by comparing calculated and theoretical RI considering the same column) was finally considered to match the validation hypothesis.  

Indeed, VOCs identified in the fermentation medium were relevant compounds found in the raw matrix or synthesized by the microorganism’s consortium, as backed up in the manuscript regarding the literature.

These aspects had been justified and presented in the Material & Methods 2.4 Chromatographic analysis of the development of volatile organic compounds manuscript section as followed:

“To optimize compound identification and data processing, the following filters were applied as a first rough compound sorting on the raw data: match factor (percentage of probability of correct identification of the compound under consideration) ≥ 70% and maximum area under the curve ≥ 0.05% (except for aroma analysis: ≥ 1%). Next, a manually disciplined treatment approach consisted in the validation of all molecules one by one, referring to the mass spectrum by consulting the spectra of reference molecules in the literature. Additionally, RI was finally considered to match the validation hypothesis [41].”

          b. OAV quantification

Response:

The determination of Odor-activity values was calculated as followed:

Where C (ppm) was the compound concentration in the considered sample (semi-quantification calculated relative to the injected internal standard) and OT its orthonasal detection odor threshold. The OT values are referred to the literature, in water.

These aspects had been justified and presented in the Material & Methods 2.4 Chromatographic analysis of the development of volatile organic compounds manuscript section as followed:

“The determination of OAV (odor-activity value) was calculated as a function of the considered compound concentration in the sample divided by its orthonasal detection perception threshold in water, referred in the literature [42]. This concept allowed to study the contribution of the compounds to the aroma of the drink [43].”

  1. Concern regarding the VOCs identification in the 3.2.1 Chemical analysis of major VOCs and metabolic pathways involved in fermentation.

Response:

Thank you for your pertinent comment aimed at improving my work.

Within our research unit and our laboratories, we have a large database allowing the identification of RIs according to the type of chromatographic column used in the experiments. As described in the first concern’s response, all raw data were manually treated to obtain a rigorous VOCs dataset validated with the literature.

When compiling the relevant dataset into the output file corresponding to Table 1 showing the identified VOCs and their characteristics, I realized that I had imported the RI validation from another chromatographic column (HP-5MS). This is why the RI difference seemed significantly different. However, the VOCs were carefully processed and identified in the raw file, validating them to the literature against the same capillary polar column used in the laboratory (Carbowax 20M).

A complete verification of calculated RI (determined by n-alkanes, Kovats method) and theoretical RI (reported in the literature for capillary polar Carbowax 20M column) has been implemented on the complete VOCs database. Indeed, a ΔRI ≤ 10 is now assured, all compounds considered.

I am deeply sorry for this inconvenience and thank you for raising this point.

This corrected section (3.2.1) based on your remarks is available in the uploaded file, highlighted by the modification marks mode.

    3. Concern regarding the consideration of an extra temperature modality in the analysis

Response:

Thank you for your pertinent comment aimed at improving my work.

All the experiments were carried out according to two modalities of the "fermentation temperature" factor, which are 20 and 30°C. The choice to present and discuss only the results obtained under the 30°C temperature modality was made considering the similar trends observed in the results between the 2 temperatures implemented. Thus, physico-chemical results obtained at 20°C are data not shown. Moreover, the 30°C temperature modality induced the most optimal fermentation from the various organic acid’s development point of view, as backed up with previous studies conducted on kombucha beverages. This temperature modality selection was then decided in a way of presenting the most relevant information in this manuscript.

However, as fermentation temperature parameter might induce variations in the VOCs synthesis and ultimately on the aroma profile of the drink, an insight on the temperature impact is proposed in this manuscript (by considering the modality of 20°C), regarding the 3.2.2 Aromatic characterization of kombucha and establishment of sensory profiles section. This trait allows the qualitative comparison of the aroma active-compounds development. These discussion tracks allowed to consider the temperature parameter as another way of controlling fermentation flavors, not to mention that major fermentation modalities may be linked with each other (fermentation time for instance).

These aspects had been modified in the 3. Results and Discussions manuscript section as followed:

“The set of experiments carried out in this study included the fermentation of kombucha at two different temperatures: 20 and 30°C. The choice to present and discuss only the results obtained under the 30°C temperature modality was made considering the similar trends observed in the results between the 2 temperatures implemented. Thus, physico-chemical results obtained at 20°C are data not shown. Moreover, 30°C temperature modality induced the most optimal fermentation from the various organic acids development point of view [24,29,30]. However, as fermentation temperature parameter might induce variations in the VOCs synthesis and ultimately on the aroma profile of the drink, an insight on the temperature impact is proposed in this manuscript (by considering the modality of 20°C), regarding the 3.2.2 section.

          4. Concern regarding the statistical significative results for the 1.2. Carbohydrates, Alcohol and Acids concentration kinetics during kombucha fermentation manuscript section

Response:

Thank you for your remark of high relevance and interest.

For each result highlighted and discussed in the 3.1.2 manuscript section, a Student’s t test (n=3, α= 0.05) was performed on the relative raw data. As a result, the significative difference observed (if any) between the two considered compound concentrations fermentation stages was supported by the t test p-value characteristic and its signification.

Indeed, I backed up my words and specified the degree of significant difference regarding components concentrations stages.

This corrected section (3.1.2) based on your remarks is available in the uploaded file, highlighted by the modification marks mode.

Moreover, Student’s t test parameters employed to assess statistical statements from this section were mentioned in the Material and Methods 2.5 Raw data analysis manuscript section.

These aspects had been modified in the Material & Methods 2.5 Raw data analysis manuscript section as followed:

Student's t-tests were performed on the pH and the spectrophotometry data to determine if there were significant differences (n=3; α = 5%) measured between the stages of fermentation.”

I hope that my explanations and responses to your comments would meet your expectations. 

Best regards, 

Suffys Sarah

Reviewer 3 Report

Comments and Suggestions for Authors

Dear Authors,

In general, the manuscript is good to read, the structure of the work is clear. After reading the paper, it is clear that the authors performed a lot of experiments and analyzes in order to obtain detailed research results. I present my comments below:

1. “2.1. Materials and chemical reagents” - What is the rationale behind the selection of such ingredients. This needs to be clarified.

2. "2.2. Kombucha preparation" - Is this the standard for kombucha preparation? Is it the most popular method of preparation? Please comment and explain.

3. Why did the authors choose such a sampling interval? Every two days and in two cases every 3 days?

4. Line 154. Is there any justification for choosing such a column?

5. Figures 5a and 5b are difficult to read. Does the second principal component describe the changes between the first stage of fermentation T0-T2 and the second stage T3-T6?

6. The "Conclusion and prospects" section needs redrafting. First of all, it's too long. Still, it's better for the reader to have the main conclusions in the "Conclusion" section. As it stands, the section is more like a "Results Discussion" rather than a "Conclusion". For example, paragraph 473-478, or 481-482 etc...

Should there be references to citations in this section?

Author Response

Dear Reviewer,

I would like to thank you for the proofreading of my article and the relevant comments and remarks established, contributing to the improvement of my work.

Please find enclosed my response relative to the concerns you mentionned in this report. 

Hoping that this explanation would meet your expectations.

Best regards,

Suffys Sarah

Reviewer 4 Report

Comments and Suggestions for Authors

The paper “Characterization of aroma active compounds production during kombucha fermentation: towards the control of sensory profiles is focused on assessing the main volatile compounds and the odor-active compounds from kombucha drinks to estimate consumer perception.

The paper contains relevant information. Adequate analytical methods along with relevant statistical analysis applied to the raw data allowed the authors to obtain useful conclusions for their study. 

However, some observations are:

  1. For the Introduction section, I suggest a more clear statement of the paper’s aim. 
  2. Also, since fermentation temperature is one of the experimental factors of the study, a short critical point of view of its influence should be presented in the Introduction section.  
  3. Materials and methods section 

-line 127, please indicate the producer, city, and country for the pH-meter

-line 132, the same observation for the spectrophotometric analysis

3. Results and discussion section

- line 186, „.. at 30°C: the temperature that probably induces the most optimal fermentation….”; this statement must be more clearer and more expressed by taking into account the other factors involved in the process 

-line 204, please define abbreviation at first use (MOs); please verify in all manuscript this aspect

-since the table's caption must be self-explanatory, I suggest completing the Table 1 title by indicating the temperature of 30â—¦C.

-same observation for Table 2 and also, please indicate the databases used for the aroma description category.

The main issue of this section is the lack of discussion regarding the precursors, mechanisms etc. of the aroma compounds formation during fermentation. The raw matrix, the microbial consortium, the fermentation factors all contributed to the formation of the volatile derivatives an is important to explain their influences.  A critical overview of these aspects must be introduced in the manuscript. 

4.The Conclusion section is too long. It must highlight only the essential conclusions.

Author Response

(The authors gave the same response as above.)

Round 2

Reviewer 4 Report

Comments and Suggestions for Authors

I agree with this form of the paper.